# A Transfected *Babesia bovis* Parasite Line Expressing eGFP Is Able to Complete the Full Life Cycle of the Parasite in Mammalian and Tick Hosts

**DOI:** 10.3390/pathogens11060623

**Published:** 2022-05-27

**Authors:** Wendell C. Johnson, Hala E. Hussein, Janaina Capelli-Peixoto, Jacob M. Laughery, Naomi S. Taus, Carlos E. Suarez, Massaro W. Ueti

**Affiliations:** 1Animal Diseases Research Unit, USDA-ARS, Pullman, WA 99164, USA; carl.johnson@usda.gov (W.C.J.); naomi.taus@usda.gov (N.S.T.); carlos.suarez@usda.gov (C.E.S.); 2Program in Vector-Borne Diseases, Department of Veterinary Microbiology and Pathology, Washington State University, Pullman, WA 99164, USA; hala.elsayed@wsu.edu (H.E.H.); j.capellipeixoto@wsu.edu (J.C.-P.); j.laughery@wsu.edu (J.M.L.); 3Department of Entomology, Faculty of Science, Cairo University, Giza 12613, Egypt; 4Paul G. Allen School for Global Animal Health, Washington State University, Pullman, WA 99164, USA

**Keywords:** *Babesia bovis*, *Rhipicephalus microplus*, transmission, transfected parasite, whole gene replacement

## Abstract

Bovine babesiosis is caused by apicomplexan pathogens of the genus *Babesia*, including *B. bovis*. This protozoan parasite has a complex life cycle involving dynamic changes to its transcriptome during the transition between the invertebrate and vertebrate hosts. Studying the role of genes upregulated by tick stage parasites has been hindered by the lack of appropriate tools to study parasite gene products in the invertebrate host. Herein, we present tfBbo5480, a transfected *B. bovis* cell line, constitutively expressing enhanced green fluorescent protein (eGFP) created by a whole gene replacement transfection strategy, that was capable of completing the parasite’s entire life cycle in both the vertebrate and invertebrate hosts. tfBbo5480 was demonstrated to respond to in vitro sexual stage induction and upon acquisition by the female tick vector, *Rhipicephalus microplus*, the tick specific kinete stage of tfBbo5480 was detected in tick hemolymph. Larvae from tfBbo5480 exposed *R. microplus* female ticks successfully transmitted the transfected parasite to a naïve calf. The development of the whole gene replacement strategy will permit a deeper understanding of the biology of parasite-host-vector triad interactions and facilitate the evaluation of upregulated genes during the parasite’s journey through the tick vector leading to new intervention strategies for the control of bovine babesiosis.

## 1. Introduction

Bovine babesiosis is one of the most important tick-borne intraerythrocytic protozoal parasite diseases of livestock in tropical and subtropical areas worldwide [1,2]. The disease is caused by *Babesia bovis*, *B. bigemina*, and *B. divergens* which are transmitted exclusively by ixodid ticks, such as *Rhipicephalus microplus*. Acute babesiosis is characterized by high fever, parasitemia, hemolytic anemia, hemoglobinuria, anorexia, and, in many cases associated with *B. bovis*, neurological signs leading to the death of the mammalian host [3,4]. Infection of vertebrate and invertebrate hosts is required for *Babesia* to complete its complex life cycle [5]. In the mammalian host erythrocyte, *Babesia* parasites undergo binary replication of haploid stages, exit from the infected cell by lysis, and invade new erythrocytes. In contrast, in the tick vector, parasites utilize sexual reproduction with diploid stage development and new morphological forms. In the lumen of the tick midgut, ingested blood stage parasites differentiate into extracellular sexual stages that fuse into zygotes. Zygotes invade midgut epithelial cells, and mature into elongated kinete forms which are subsequently released into the tick hemocoel, thereby gaining access to various organs, including the ovary and eggs resulting in vertical transmission of the parasite to the tick offspring [6]. In tick salivary gland acinar cells of infected larvae, another morphological change occurs as parasites develop into sporozoites that are eventually inoculated into the bovine host via tick saliva to begin a new cycle of erythrocyte invasion [3,7].

The complexity of the *Babesia* life cycle has hindered the understanding of the basic biology of the parasite. Since the development of the microaerophilous stationary phase (MASP) in vitro culture system for *Babesia* [8], knowledge regarding the parasite’s blood stages has been greatly expanded with the identification of parasite surface proteins as potential vaccine targets [9], sensitivity to a variety of drug therapies [10,11], and, importantly, the development of CRISPR/Cas9 and transfection systems to manipulate the parasite’s genome [12,13,14]. The transfection system inserts DNA sequences into one of the two copies of Elongation Factor 1 alpha (EF1α) and has been used to express foreign proteins including green fluorescent protein [13], *R. microplus* BM86, or tick glutathione-S-transferase by transfected *B. bovis* during infection of the mammalian host [15], and to explore the mechanism of parasite erythrocyte invasion [16]. However, insertion into the locus effectively knocks out one copy of the EF1α gene with documented effects on the cell cycle growth characteristics of similarly transfected *Plasmodium falciparum* [17]. Herein, we describe an alternative transfection strategy that replaces the whole gene with new genetic sequences that retain, or potentially modify gene function. Replacement occurred without the consequences of gene knock out and resulted in a transfected parasite that was capable of infecting both the vertebrate and invertebrate hosts culminating with the successful transmission of the transfected *B. bovis* to a naïve calf by infected larvae.

## 2. Results

### 2.1. Transfection of Parasites

Freshly collected ex vivo *B. bovis* were cultivated in vitro for five days until the parasitemia reached ~20%. Parasites were then electroporated with the transfection plasmid pBS-5480 KI (schematically represented in Figure 1A) and cultured under blasticidin selection for approximately one week at 37 °C with 5% CO_2_. Using epifluorescence microscopy, green fluorescing tfBbo5480 was detected inside infected erythrocytes (Figure 1B). Erythrocytes containing Hoechst-stained parasites that did not fluoresce for eGFP indicated the presence of non-transfected wild type (WT) parasites. The transfected culture was maintained with blasticidin for 52 days to select for eGFP expressing *Babesia* parasites. However, Hoechst-stained non-eGFP fluorescing parasites were still detectable, suggesting the presence of a mixed population of transfected and non-transfected parasites remained in the in vitro cultured line (Figure 1B circle).

### 2.2. In Vitro Babesia Sexual Stage Induction

Previous work demonstrated that induction of sexual stages in cultured *B. bovis* yielded two extracellular morphological forms, a round form and a form with appendages that aggregate during culture [18]. Interestingly, tfBbo5480 induced cultures contained such fluorescent sexual forms, when viewed by epifluorescence. As expected, non-transfected parental S74-T3Bo parasites used as a control also transitioned into non-fluorescent sexual forms upon induction, that were morphologically indistinguishable as compared to parasites of the transfected line tfBbo5480. In both the transfected and WT *B. bovis* induced cultures the sexual stages exhibited an aggregation behavior (Figure 2). Overall, the results indicate that transfected parasites successfully transitioned into sexual forms in a pattern that was similar to non-transfected parental parasites.

### 2.3. Parasite Infection and during Tick Acquisition Feeding

Calves inoculated with tfBbo5480 or WT *B. bovis* developed acute *Babesia* infection, including temperatures exceeding 40 °C, anorexia, and a reduction in PCV of at least 20%. From the calf infected with tfBbo5480, circulating erythrocytes infected with green fluorescing *Babesia* parasites were detected by day seven post-inoculation (Figure 3, top panel). From the calf infected with WT *B. bovis*, erythrocytes infected with parasites were detected only by light microscopy on day 10 post-inoculation. No evidence of eGFP expression (Figure 3 bottom panel) by blood stage WT parasites was found. *Rhipicephalus microplus* female ticks acquired *Babesia* parasites during peak parasitemia.

### 2.4. Microscopic Detection of tfBbo5480 Kinetes Expressing eGFP

Hemolymph was collected from replete female ticks incubated for eight days. Epifluorescence microscopy of hemolymph from tfBbo5480 exposed ticks revealed *B. bovis* eGFP expressing kinete stage parasites (Figure 4 top panel). In contrast, kinetes from ticks fed on the calf infected with WT *B. bovis* were only detected by light microscopy of Giemsa-stained slides or epifluorescence of nuclei in Hoechst 33342 stained cells. There was no evidence of a green fluorescence signal (Figure 4 bottom panel) by WT *B. bovis* kinetes from the hemolymph.

### 2.5. Tick Transmission Feeding Experiment

Larvae infected with tfBbo5480 transmitted parasites to a naïve calf (C1674) and erythrocytes infected with *B. bovis* parasites expressing eGFP were detected at day 14 post-infestation (Figure 5, top panel). Larvae infected with non-transfected WT *B. bovis* also transmitted parasites to a naïve calf (C1777), and erythrocytes infected with non-fluorescent parasites were observed on day seven (Figure 5 bottom panel). 

### 2.6. In Vitro Culture Recovery and Integration, Analysis of tfBbo5480 Parasites from Infected Calves

Circulating *B. bovis* parasites were successfully recovered in in vitro cultures from calves used in tick acquisition (Figure 3) and transmission (Figure 5) feeding experiments. Analysis of these cultured parasites showed green fluorescing parasites when analyzed by epifluorescence microscopy. Primers, designed to amplify inside the cassette and up- and down-stream of genomic *B. bovis* BBOV_II005480 guide sequences, amplified identical PCR products (Appendix A) when applied to DNA from both recovered parasite lines (Figure 6). Sequencing of both amplicons generated from tfBbo5480 acquisition and transmission calves confirmed that acquisition and transmission fed calves were infected with tfBbo5480. In contrast, parasites recovered from WT *B. bovis* acquisition and transmission fed control calves did not amplify such amplicons. Integration of the transfection cassette into chromosome 2 by homologous recombination was demonstrated by PCR Reactions 1–3, showing that the cassette was stably inserted and retained throughout tick transmission (Figure 6A). PCR 1 and 2 used primers outside the guide regions and therefore spanned from unmodified genomic sequences into the insert.

The identical sequenced amplicons (Figure 6B) indicated that the replacement of native BBOV_II005480 by the modified gene remained intact throughout sexual reproduction in the female tick and subsequent larval transmission. PCR spanning from inside the gene to the 3′ flanking region of genomic DNA demonstrated that two genotypes were present in the tfBbo5480 cell line with PCR products (Appendix A) predicted for both WT and transfected parasites DNA (Figure 7A). Again, both genotypes were able to develop their full life cycle through the vertebrate and tick vector hosts and were transmitted by infected tick larvae. 

### 2.7. Expression Analysis of tfBbo5480 Parasites from Infected Calves

Reverse transcriptase PCR showed that mRNA was made from the newly inserted DNA. Gene products for both the modified replaced gene, including the appended DNA sequence (Appendix A), and eGFP-BSD DNA (Appendix A) amplified from the samples from tfBbo5480 infected calves but not control WT *B. bovis* infections (Figure 7B1,B2). RT-PCR of the unmodified portion of the replaced gene (Figure 7B3) identified both the modified and native genes in all samples tested (Appendix A). Similarly, RT-PCR for the *B. bovis* gene *rap-1* identified all samples as being infected with the parasite (Figure 7B4) (Appendix A). Altogether, the expression and genetic analysis data confirmed that the transfected genes remained stable during the process of sexual reproduction and tick stage development.

## 3. Discussion

Despite the remarkable advances in studying *Babesia* parasite stages of the vertebrate host both in vitro and ex vivo, there are limited studies examining the interaction between parasites and cells of the invertebrate host. Recently, in vitro induction of *Babesia* sexual stages has been used to demonstrate stage specific expression of a variety of proteins such as HAP2, 6-Cys-B, CCP2, and CCP3 [18,19,20] typically expressed by gametes in the tick midgut lumen. In addition, a recent RNA-Seq approach revealed an extensive turnover of the transcriptome between the blood stage merozoite and tick stage kinete forms of *B. bovis* [21], indicative of the parasite adapting to new microenvironments in the invertebrate host. However, little is known regarding the function of the genes upregulated during infection of the tick and how their gene products are used by the parasite to interact with tick cells during its migration from the midgut lumen to the infection of tick progeny.

To address this fundamental knowledge gap, we proposed to use the *Babesia* transfection system to probe the function of proteins upregulated during the kinete stage. However, to date, there are no descriptions of transfected *B. bovis* successfully being transmitted through the tick vector, *R. microplus*, and attempts to transmit transfected *B. bovis* through the vector to naïve bovines have failed (personal communication, Carlos E. Suarez). Transfection of *Plasmodium* sp also targets EF1α and has been used successfully to transmit transfected parasites throughout the parasite’s entire life cycle, including development within the mosquito vector [17]. However, the development of sexual stages of *Plasmodium* begins within the mammalian host, whereas *Babesia* gametes develop within the lumen of the tick midgut [22,23]. Importantly, transfection into the *Plasmodium* EF1α locus has been shown to alter the growth cycle characteristics of transfected parasites, extending the duration of the G1 phase [17]. We speculated that alteration to the timing of cellular developmental events may jeopardize the ability of EF1α-targeted transfected *Babesia* to correctly develop within the tick vector. To circumvent this issue, we developed a novel whole gene replacement strategy targeting the alternative locus tag BBOV_II005480, an AP2 DNA transcription factor previously identified as highly expressed by blood stages but down regulated in the kinete stage [21]. The data suggest that our strategy resulted in a tick transmissible transfected parasite line that replaced the WT gene sequence with a genetically modified form of the gene and also expressed the transfected eGFP-BSD gene.

One reason we selected BBOV_II005480 as a transfection target was that this *Babesia* AP2 shares identity with the DNA transcription factor designated AP2-G in *Plasmodium* due to its involvement in gametogenesis. If the protein serves a similar role in *Babesia* as previously suggested [24], then in vitro induction of sexual stages would be an indication that transfection did or did not compromise that function. We demonstrated that tfBbo5480 responded normally to in vitro sexual stage induction producing two fluorescent sexual forms that aggregated as previously described [18,25], suggesting that normal sexual stage development of the transfected parasite would occur in the lumen of the tick midgut. Another important consideration was that in *Plasmodium*, development of sexual forms was compromised by the duration of in vitro culture [26]. Recently, the failure of long-term in vitro cultivated *B. bovis* to develop sexual stages upon XA induction was reported, and, in addition, the long-term in vitro cultivated *B. bovis* was non-tick transmissible [27]. To prevent the loss of the sexual stage induction response and maintain tick transmissibility, we transfected a freshly isolated *B. bovis* strain isolated from calves infected with tick transmissible parasites and minimized the time spent under blasticidin selective pressure. The use of blasticidin retards the growth of non-transfected parasites [13]. The result was a transfected cell line that retained some WT *B. bovis*. We took advantage of the presence of WT parasites in the transfected parasite line to serve as an internal control to demonstrate that tick transmissibility was not compromised by the relatively short-term culture of tfBbo5480. As evidence of this fact, PCR that spanned from the 3′ end of the gene to the intergenic region of genomic DNA from both the acquisition and transmission infections showed that two genetically distinct parasite populations were present prior to tick acquisition of the transfected cell line and after tick transmission. Additionally, RT-PCR results showed that the replaced gene with modification was correctly transcribed by blood stage parasites. Further experiments using tfBbo5480 from progressively longer culture times will allow the assessment of when tick transmissibility is lost.

Previously, co-infection with tick transmissible and non-transmissible *B. bovis* strains allowed for both strains to be transmitted by the tick vector [28]. However, the mechanism to regain transmission was not defined. Several possibilities were postulated, including recombination of *B. bovis* parasites during sexual fusion and production of enzymes by the transmissible strains that facilitated the non-tick transmissible strain to infect tick midgut. In this study, a mixed population of *B. bovis*, tfBbo5480 and non-transfected parasites, was used. Both populations were transmitted by *R. microplus* to naïve animals. However, it is unclear if the transfected parasite alone remains tick transmissible. One alternative strategy to elucidate this question would be to purify transfected parasites using Fluorescent Activated Cell Sorting technology to obtain a transfected parasite line that is free of non-transfected WT *B. bovis*.

A distinguishing characteristic between the described EF1α transfection approach, which knocks out the targeted gene, and the whole gene replacement strategy is that gene expression of the targeted gene should not be affected. Reverse transcription PCR using cDNA synthesized from mRNA isolated from both the acquisition and transmission calves demonstrated that the modified gene was, in fact, inserted and replaced the WT gene with a new novel sequence which was successfully transcribed by tfBbo5480. While we used 700 bp sequences at the 5′ and 3′ end of the gene to be replaced, we believe that the gene itself could have been used as the 5′ guide, thereby inserting new genes into the trailing 3′ intergenic region. This technique will facilitate the study of both newly inserted intergenic and modified parasite gene products inside the invertebrate host. The whole gene replacement technique described herein could be used to evaluate the effect of gene modification on protein function within the tick vector and its effect on tick transmissibility, assess kinete specific gene promoters, as well as deliver novel gene sequences into the genome that are expressed exclusively by parasite tick stages. Additionally, fluorescent parasite forms will aid in tracking the parasite into the larval stage of the tick, providing opportunities to study the as-yet poorly characterized sporozoite stage of *B. bovis*.

## 4. Materials and Methods

### 4.1. Calves, Tick-Vector, and Parasites

Holstein calves approximately 4 months of age and determined to be *Babesia*-free by RAP-1 competitive enzyme-linked immunosorbent assay and nested *rap-1*-PCR [6,7], were splenectomized and used for *R. microplus* tick acquisition and transmission of WT *B. bovis* or transfected parasites. The *R. microplus* La Minita tick strain, a competent vector for *B. bovis* [29], was employed in this study. *Babesia bovis* S74-T3Bo Texas strain was utilized to develop a transgenic parasite expressing enhanced green fluorescent protein (eGFP) by homologous recombination.

### 4.2. Transfection Cassette Construction

The transfection plasmid cassette (Figure 1 panel A) contained a full-length copy of BBOV_II005480 appended with a unique 108 bp (Appendix A) identifying sequence before the stop codon, 3′ *rap-1* stop, elongation factor promoter B intergenic region controlling the expression of the eGFP gene fused with the blasticidin S deaminase gene (eGFP-BSD), and 700 bp sequences from the 5′ and 3′ untranslated regions flanking BBOV_II005480 as guides to be inserted into the BBOV_II005480 locus by homologous recombination. The unique 108 bp sequence, which codes for 36 amino acids, was appended to allow discrimination between transfected and wildtype mRNA. The transfection cassette was directly inserted into the pBluescript plasmid and synthesized (Genscript Biotech, Corp., Piscataway, NJ, USA). The plasmid was transformed into *Escherichia coli* and isolated with a Qiagen endotoxin-free maxiprep kit (Qiagen Inc., Redwood City, CA, USA) following the manufacturer’s instructions.

### 4.3. Transfection of Parasites

Wild type *B. bovis* Texas strain stabilate was inoculated into a splenectomized calf (C1660). During acute parasitemia, blood samples were collected from the infected calf into flasks containing glass beads and shaken to prevent blood coagulation. Defibrinated blood samples were washed with Puck’s Saline G, placed into flasks with a culture medium as previously described [8,21], and incubated in vitro with 5% CO_2_ for five days to increase the percentage of *B. bovis*-infected erythrocytes prior to transfection. When the parasitemia reached ~20%, *Babesia* parasites were electroporated with 0.5 μg of plasmid cassette as previously described [13]. Transfected *B. bovis* were cultured in a medium containing 4 μg/mL of blasticidin (Thermo Fisher Scientific, Waltham, MA, USA) with a packed cell volume (PCV) of 10% and incubated at 37 °C with 5% CO_2_. Blasticidin-resistant parasites that emerged expressing eGFP were visualized using epifluorescence microscopy and termed tfBbo5480 to identify the cell line as a transfected replacement of gene BBOV_II005480. To determine if the homologous recombination occurred specifically in the targeted locus on chromosome 2, PCR primers targeting inside the cassette and outside of the inserted guide 5′ and 3′ sequences for BBOV_II005480 were designed (Table 1) and amplicons sequenced (Eurofins MWG Operon, Louisville, KY, USA).

### 4.4. In Vitro Sexual Stage Induction

To induce sexual stages from tfBbo5480 and WT *B. bovis*, stabilates collected from infected calves were recovered and maintained in in vitro cultures for two weeks before induction. tfBbo5480 and WT *B. bovis* cultured infected erythrocytes were suspended in a medium with or without 100 μM xanthurenic acid (XA) (Sigma, St. Louis, MO, USA) as previously described [18,25]. Induced sexual stage live parasites were collected at 24 h post-induction. To visualize the gamete formation, parasite cells were incubated with the nucleic acid stain Hoechst 33342 (Thermo Fisher Scientific) for 30 min and washed twice with PBS. Five µL from each sample of live parasites was used per slide. Slides were examined and visualized independently by fluorescent microscopy with eGFP and DAPI channels using a Leica microscope using LAS-X software.

### 4.5. Detection of Transfected Blood Stage and Kinete Stage Parasites Expressing eGFP 

Approximately 40,000 *R. microplus* larvae were placed under a cloth patch on a splenectomized calf (C1667) and allowed to feed through to adult repletion. At 14 days post-larval application, when the ticks had molted to the adult stage, the splenectomized calf was intravenously inoculated with ~10^7^ tfBbo5480 infected erythrocytes. Female tick repletion was synchronized to acquire parasites at peak parasitemia of transfected parasites in the peripheral blood. The presence of tfBbo5480 in calf blood was determined using wet mount slides, stained with Hoechst 33342, and examined by epifluorescence microscopy for transgenic parasites expressing eGFP as described above.

Replete female ticks were collected, rinsed in tap water, and placed individually into 24-well tissue culture plates and incubated at 26 °C and 92% relative humidity for kinete development in tick hemolymph. On day eight of incubation, hemolymph was sampled from individual ticks. A distal leg segment was removed, and a drop of exuded hemolymph was placed onto a glass slide, Giemsa stained, and examined by light microscopy as previously described [30]. Highly infected female ticks were selected and infected hemolymph containing kinetes was collected. The kinetes were added to a wet mount slide, stained with Hoechst 33342, and evaluated for live tfBbo5480 kinetes expressing eGFP by epifluorescence microscopy as described above.

For WT *B. bovis*, a splenectomized calf (C1763) was inoculated with a non-transfected Texas strain, and ticks were fed during acute infection in the same manner as above. Engorged female ticks were evaluated to detect kinetes in tick hemolymph. Hemolymph was collected as described above, and kinetes were visualized by Giemsa-stained light microscopy. For epifluorescence, hemolymph from highly infected ticks was collected, treated with Hoechst 33342, and used as a WT kinete control for autofluorescence by epifluorescence microscopy.

### 4.6. Transmission of tfBbo5480 and WT B. bovis

Egg masses from replete female ticks infected with either tfBbo5480 or WT *B. bovis* were collected and incubated at 26 °C and 92% relative humidity. Larvae from collected tfBbo5480 exposed egg masses were applied under a cloth patch on a naïve splenectomized calf (C1674) and fed for five days to transmit tfBbo5480. After terminating larval feeding using acaricides, the transmission was determined by evaluating infected blood for tfBbo5480 using light microscopy of Giemsa-stained blood smears and evaluated for eGFP expression on wet mount slides by epifluorescence microscopy. 

Larvae from female ticks fed on a control calf were applied under a cloth patch on a splenectomized calf (C1777) and fed for five days to transmit WT *B. bovis* to a naïve calf. After terminating larval feeding using topical Permectrin II (Bayer Healthcare LLC, Shawnee Mission, KS, USA), the transmission was evaluated daily for 14 days using light microscopy of Giemsa-stained blood smears and wet mount slides by epifluorescence microscopy. 

### 4.7. In Vitro Culture Recovery and Integration Analysis of tfBbo5480 Parasites from Infected Calves

Genomic DNA was isolated from all four infected calves and used for PCR. Briefly, tfBbo5480 and WT *B. bovis* short-term in vitro cultures prepared as above were expanded to 5% parasitemia. The erythrocytes were pelleted and washed with phosphate-buffered saline (PBS) (Thermo Fisher Scientific). Erythrocytes were lysed using red blood cell lysis solution (Qiagen) and incubated for 5 min at room temperature. Lysed red cells were centrifuged for 3 min at 2800× *g* and the supernatant was discarded. Parasite pellets were suspended in 40 µL of PBS and 500 µL of cell lysis solution (Qiagen) with 20 μg/mL of Proteinase K (Qiagen) and incubated at 56 °C for 30 min. Proteins were removed using Protein Precipitation Solution (Qiagen), and DNA precipitated using isopropanol, washed with 70% ethanol, and suspended in 50 μL of DNA hydration solution (Qiagen). PCR was performed from gDNA samples using primer sets (Table 1). PCR cycling conditions consisted of 95 °C for 3 min followed by 39 cycles of 95 °C for 30 s, 55 °C for 40 s, and 68 °C for 3.5 min, with a final extension of 72 °C for 5 min. PCR products were visualized by 1% agarose gel electrophoresis. PCR amplicons were cloned into PCR 2.1-TOPO^®^ (Thermo Fisher Scientific) and submitted for sequencing (Eurofins MWG Operon).

### 4.8. Expression Analysis of tfBbo5480 Parasites from Infected Calves

Total RNA was extracted from tfBbo5480 and WT parasites using TRIzol reagent (Thermo Fisher Scientific) according to the manufacturer’s protocol, and the RNA pellets were suspended in 20 µL DEPC-treated water. RNA samples were treated with DNase I (Thermo Fisher Scientific) following the manufacturer’s protocol to remove contaminating genomic DNA and quantified by Nanodrop (Thermo Fisher Scientific). The removal of genomic DNA was confirmed by PCR targeting *rap-1* as previously described [31]. cDNA was synthesized from 100 ng of total RNA from each sample with a Superscript^®^ First-strand cDNA synthesis kit (Thermo Fisher Scientific) with (RT+) and without (RT-) enzyme following the manufacturer’s protocol. 

Reverse transcription polymerase chain reactions (RT-PCR) were performed from cDNA samples using primer sets (Table 1). PCR cycling conditions consisted of 95 °C for 3 min followed by 35 cycles of 95 °C for 30 s, 55 °C for 30 s, and 72 °C for 2 min, with a final extension of 72 °C for 5 min. PCR products were visualized by 1% agarose gel electrophoresis. PCR amplicons were cloned into PCR 2.1-TOPO^®^ and submitted for sequencing (Eurofins MWG Operon).

## 5. Conclusions

We exploited the *Babesia* transfection system to generate a stably transfected parasite constitutively expressing eGFP that was capable of successfully recapitulating the whole life cycle of the parasite in both the vertebrate and invertebrate hosts. Novel sequences were appended to the replaced gene without apparent loss of native protein function. This new approach will help provide tools to explore a deeper understanding of the basic biology of parasite-vector interactions and facilitate the development of novel approaches to combat bovine babesiosis.

## Figures and Tables

**Figure 1 pathogens-11-00623-f001:**
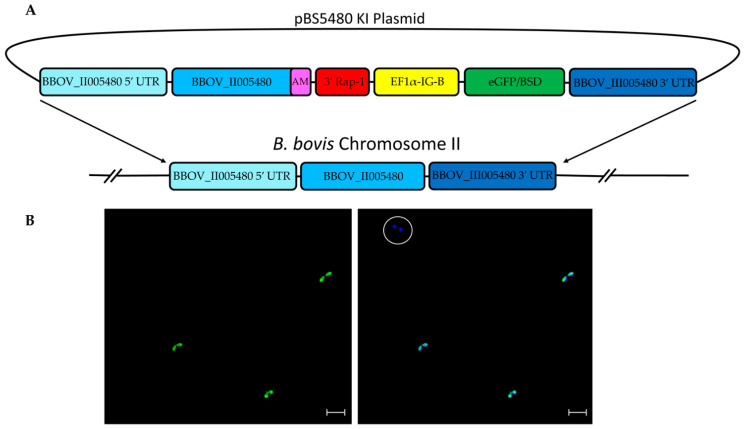
(**A**): The schematic components of the stable transfection plasmid pBS-5480 KI for insertion into chromosome 2 of *B. bovis* are shown. The dark and light blue bars represent the 5′ and 3′ 700 bp guide regions flanking the BBOV_II005480 ORF, the blue with purple bar represents the 1.47 kbp BBOV_II005480 region with appended 108 bp sequence chimera, the composite red bar represents a 3′ region of *rap-1* used to control stop of transcription, the yellow bar represents the EF1α-IG-B promoter region, and the green bar represents the egfp-bsd fusion protein. (**B**): eGFP (left) and Hoechst 33342 (right) epifluorescence images of tfBbo5480. The circle highlights a non-transfected parasite still present in the transfected cell line. Scale bar: 5 µm.

**Figure 2 pathogens-11-00623-f002:**
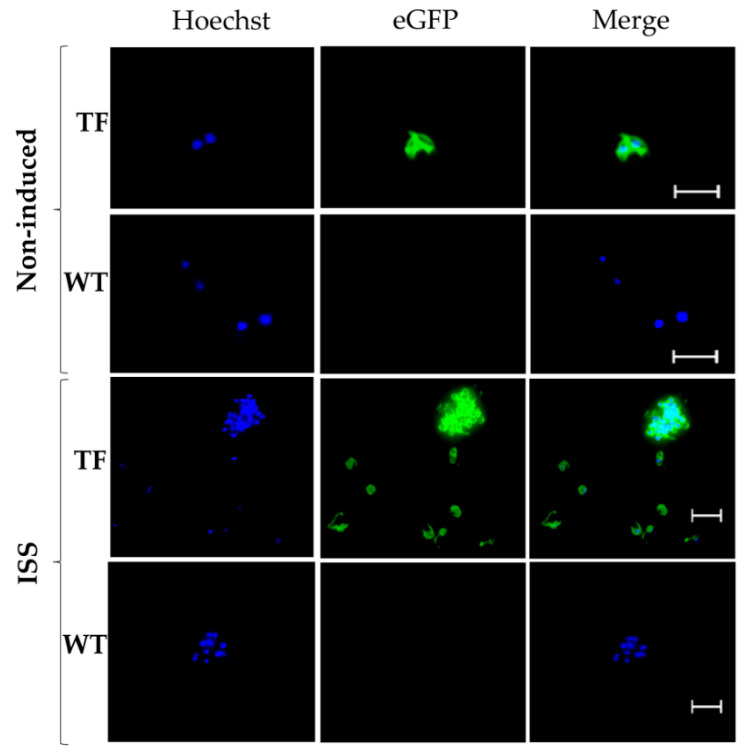
Transfected eGFP expressing parasites develop into sexual stages upon xanthurinic acid (XA) induction. tfBbo5480 (TF) and wild type *B. bovis* (WT) in vitro cultures were incubated for 24 h with (ISS) and without (non-induced) XA at 26 °C to induce sexual stages. The cells were stained with Hoechst 33342 and visualized by epifluorescence microscopy. Scale bar: 5 µm.

**Figure 3 pathogens-11-00623-f003:**
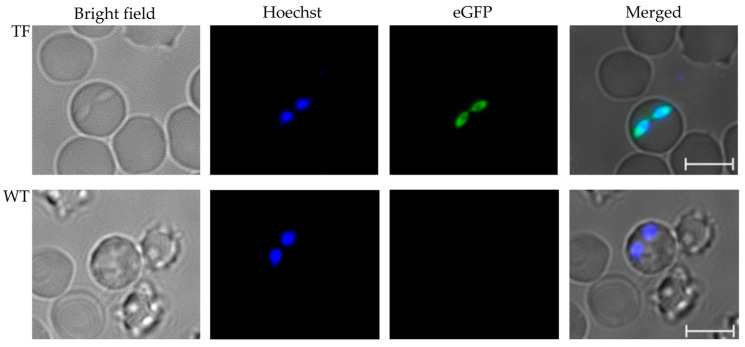
tfBbo5480 transfected parasites expressed eGFP. Ex vivo blood samples from patent calves infected with either tfBbo5480 (TF) or wild type *B. bovis* (WT) to be used for tick acquisition feeding were collected in defibrination flasks for short-term in vitro culture. Cells were stained with Hoechst 33342 and visualized by bright field and epifluorescence microscopy. Individual and merged images are presented. Scale bar: 5 µm.

**Figure 4 pathogens-11-00623-f004:**
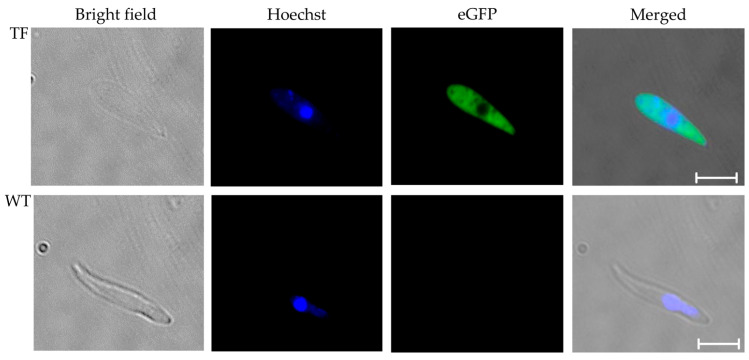
tfBbo5480 kinetes isolated from tick hemolymph expressed eGFP. *Rhipicephalus microplus* ticks were fed on calves infected with either tfBbo5480 (TF) or wild type *B. bovis* (WT). Replete females were collected and incubated at 26 °C for eight days. Hemolymph was collected from individual ticks identified as infected by Giemsa-stained slides, pooled, and stained with Hoechst 33342. Kinetes were visualized by bright field and epifluorescence microscopy. Individual and merged images are presented. Scale bar: 5 µm.

**Figure 5 pathogens-11-00623-f005:**
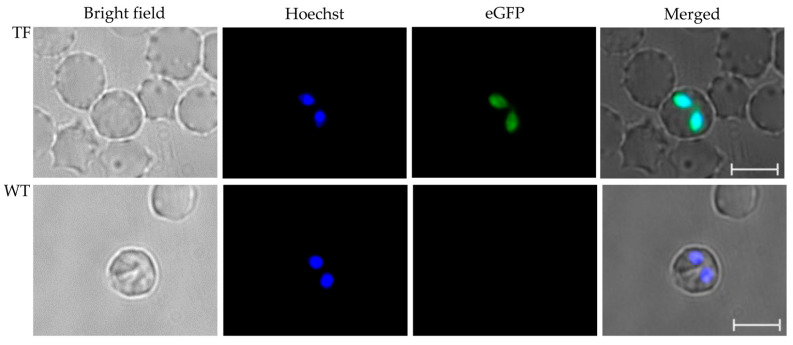
Larvae infected with tfBbo5480 transmit eGFP expressing parasites. Ex vivo blood samples from patent calves infected by exposure to larvae with either tfBbo5480 (TF) or wild type *B. bovis* (WT) were collected in defibrinating flasks for short term in vitro culture. Cells were stained with Hoechst 33342 and visualized by bright field and epifluorescence microscopy Individual and merged images are presented. Scale bar: 5 µm.

**Figure 6 pathogens-11-00623-f006:**
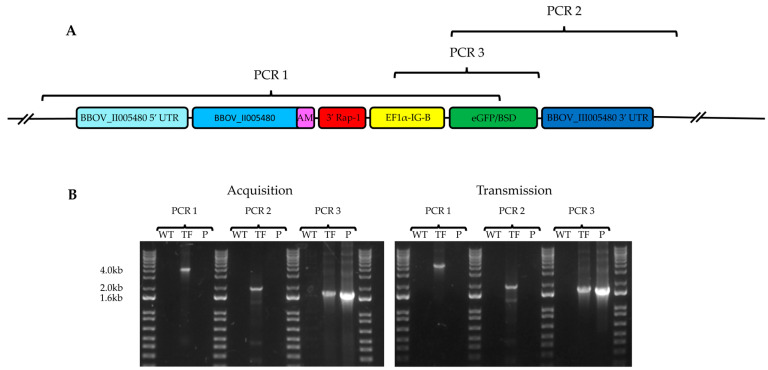
(**A**): PCR map to determine the integration of transfected elements into the *B. bovis* genome. PCR1 used a forward primer upstream of the 5′ flanking guide region and a reverse primer in the eGFP-BSD region. PCR2 used a forward primer in the eGFP-BSD region and a reverse primer downstream of the 3′ flanking guide region. PCR3 used a forward primer in the EF1α-IG-B promoter region and a reverse primer in the eGFP-BSD region. (**B**): Genomic DNA derived from calves infected with either wild type *B. bovis* (WT) or short-term in vitro cultured tfBbo5480 (TF), and plasmid pBS-5480 KI (P) (Acquisition) or genomic DNA from calves infected by larval feeding (Transmission) of either WT or TF were used for PCR and visualized in 1% agarose gels. Positive PCR1 and PCR2 reactions for both the acquisition and transmission samples were confined to the TF samples. PCR3, which amplifies within the transfection cassette, had a positive signal for both the TF and P samples.

**Figure 7 pathogens-11-00623-f007:**
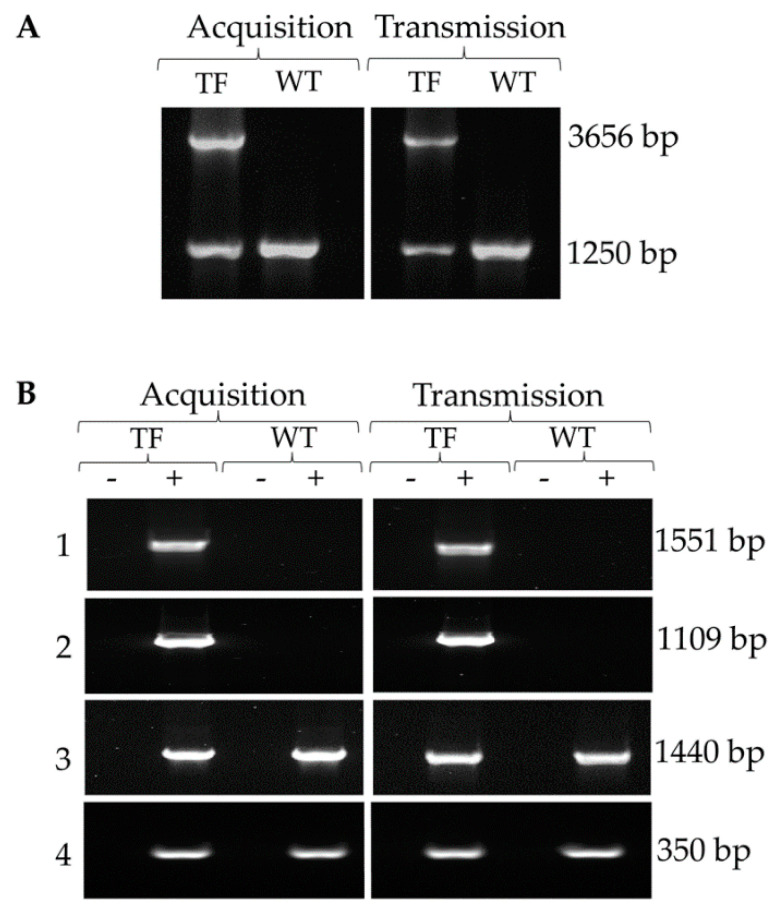
(**A**): tfBbo5480 (TF) contained transfected and non-transfected genotypes both before and after tick transmission. PCR using a forward primer beginning at position1259 bp of BBOV_II005480 and a reverse primer downstream of the 3′ flanking guide region identified two genotypes within the transfected cell line. Wild type (WT) parasites contained only the predicted 1250 base pair fragment. (**B**): RT-PCR demonstrated synthesis of mRNA by both the replaced modified gene (1) (primers BBOV_II005480-F1259 with AM-R) and inserted exogenous gene (2) (primers eGFP-F with BSD-R). Total RNA was isolated from acquisition and transmission fed short term in vitro cultures. cDNA from reverse transcriptase (RT) reactions without (-) or with (+) enzyme were used for PCR and visualized by agarose gel electrophoresis. Detection of specific gene expression was confined to RT+ samples from tfBbo5480. Controls (3) native BBOV_II005480 (primers BBOV_II005480-F1 with R1764) and (4) *Rap-1* (primers Bof with BoR) confirm the presence of *B. bovis* in both the tfBbo5480 and WT infected calves.

**Table 1 pathogens-11-00623-t001:** Primer sets used in this study to determine integration of the cassette into *B. bovis* genome.

**Integration Analysis**		
1	Forward-Upstream BBOV_II005480 GGC CAT ATT GAT ACT TGA TC	4363 bp
	Reverse-eGFP CTT GTA CAG CTC GTC CAT GC	
2	Forward-eGFP ATG GTG AGC GGC GAG GAG CTG TTC’	2034 bp
	Reverse-Downstream BBOV_II005480 CAG CAC CCC AAA TAA CTG	
3	Forward-Elongation Factor CTG GCA AAG CTT ACT TGA TCA GAT TTA	1716 bp
	Reverse-BSD GCC CTC CCA CAC ATA ACC AGA GGG CAG C	
**Transfected vs Wild Type Analysis**		
4	Forward-BBOV_II005480-F1259 ATC GCT TGG ACC TAT GTT ATG G	wt = 1250 bp tfBbo5480 = 3656 bp
	Reverse-Downstream BBOV_II005480 CAG CAC CCC AAA TAA CTG	
5	Forward-BBOV_II005480-F1 ATG TCA CTG ACT GCT CAG C	1551 bp
	Reverse–AM-R CTA GTA AGC CTG AGA GAG TAT C	
6	Forward-eGFP ATG GTG AGC AAG GGC GAG	1109 bp
	Reverse–BSD-R CCC TCC CAC ACA TAA CCA GAG GGC	
7	Forward-BBOV_II005480-F1 ATG TCA CTG ACT GCT CAG C	1440 bp
	Reverse-BBOV_II005480-R 1764 ATC CAA AAC ACT ATC ACC ATC	
8	Forward-RAP-1 (BoF) CAC GAG GAA GGA ACT ACC GAT GTT	350 bp
	Reverse-RAP-1 (BoR) CCA AGG AGC TTC AAC GTA CGA GGT CA	

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
