# Peer review of "A Transfected Babesia bovis Parasite Line Expressing eGFP Is Able to Complete the Full Life Cycle of the Parasite in Mammalian and Tick Hosts"

_pathogens, 2022, doi:10.3390/pathogens11060623_

Round 1

Reviewer 1 Report

The manuscript written by Wendell C. Johnson et al., is a excellent paper about a Babesia transfection system to generate a stably transfected parasite constitutively expressing enhanced green fluorescent protein and it capable of successfully recapitulating the whole life cycle of the parasite in both the vertebrate and invertebrate hosts. This tool could facilitate a deeper understanding of the biology of parasite-host-vector triad interactions in the future.

The topic is certainly of interest to the journal's readers and the scientific community.

The text is wery well written.

I have only a few minor observations that I would like the authors to take into account:

- In the caption of figure 1 the scale bar is absent.

- Panel A of Figure 1 should be enlarged. The panel is small and does not look good.

- It would be possible to enlarge the image in Figure 2  to visualize with more detail the two extracellular morphological forms of B. bovis.

-In the line 399 where it says "4.7. Expression..."  should read "4.8. Expression...."

Author Response

I have only a few minor observations that I would like the authors to take into account:

- In the caption of figure 1 the scale bar is absent.

Response: We added the scale bar in the figure legend.

- Panel A of Figure 1 should be enlarged. The panel is small and does not look good.

Response: We enlarged Figure 1A as recommended.

- It would be possible to enlarge the image in Figure 2  to visualize with more detail the two extracellular morphological forms of B. bovis.

Response: Our intention was to show that transfected sexual stages displayed aggregation behavior and expressed eGFP.

-In the line 399 where it says "4.7. Expression..."  should read "4.8. Expression...."

Response: We corrected as suggested.

Reviewer 2 Report

The present study aimed to investigate a transfected Babesia bovis parasite line expressing eGFP, evaluating  the full life cycle of the B. bovis. This study represent a advance of understanding of the biology of parasite-host-vector interactions, and to contribute new intervention strategies for the control of bovine babesiosis. The Babesia transfection system was capable of successfully to inspect the whole life cycle of the parasite in both the vertebrate and invertebrate hosts. 

Author Response

No comments or questions from reviewer #2.

Reviewer 3 Report

Introduction

Line 67: Explain briefly the novelty of the current approach of gene replacement.

Results

Lines 72, 93 ,100 and elsewhere: B. bovis should be written in italics.

Line 109: Describe symptoms of infected calves and if there were any differences on this aspect between the transfected vs. the control infected bovines.

Figure 6 Panel B, Transmission photo: lane 2 of PCR 3 should be “TF” instead of “KI”

Lines 139 and 142: “day 14” and “day seven” were used. Please use the same format.

Line 199: messenger RNA?

Discussion

Line 257: Explain why even under blasticidin selection, the WT parasites could still be found. Have the authors analyzed the proportions of TF and WT parasites in mixed samples?

Materials and Methods

Line 299: please address which ELISA and PCR reactions were used in this work.

Line 306: Why did the authors add this sequence? What does the appended chimera code for? Is it a short ORF, a tag?. Please explain in the text.

Section 4.6. At what times after larval feeding did the authors verify the presence of parasites in the calves.? What acaricide was used and how was it applied?. Explain in the text.

Author Response

Introduction

Line 67: Explain briefly the novelty of the current approach of gene replacement.

Response: We modified as suggested.

Lines 67-72: “Herein, we describe an alternative transfection strategy that replaces the whole gene with new genetic sequences that retain, or potentially modifies, gene function.  Replacement occurred without the consequences of gene knock out and resulted in a transfected parasite that was capable of infecting both the vertebrate and invertebrate hosts culminating with the successful transmission of the transfected B. bovis to a naïve calf by infected larvae”

Results

Lines 72, 93 ,100 and elsewhere: B. bovis should be written in italics.

Response: We corrected throughout the manuscript.

Line 109: Describe symptoms of infected calves and if there were any differences on this aspect between the transfected vs. the control infected bovines.

Response: We described the symptoms of bovine babesiosis.

Lines 114- 115:  Calves inoculated with tfBbo5480 or WT B. bovis developed acute Babesia infection, including temperatures exceeding 40oC, anorexia, and a reduction in PCV of at least 20%.

Figure 6 Panel B, Transmission photo: lane 2 of PCR 3 should be “TF” instead of “KI”

Response: We corrected as requested.

Lines 139 and 142: “day 14” and “day seven” were used. Please use the same format.

Response:  Per Grammarly, in scientific and technical writing, the prevailing style is to write out numbers under 10.

Line 199: messenger RNA?

Response:  We modified as requested.

Line 212-213: “Reverse transcriptase PCR showed that mRNA was made from the newly inserted DNA.” 

Discussion

Line 257: Explain why even under blasticidin selection, the WT parasites could still be found. Have the authors analyzed the proportions of TF and WT parasites in mixed samples?

Response:  We added a statement regarding blasticidin selection. We did not assess the proportion of TF and WT parasites.

Lines 274-225: “The use of blasticidin retards the growth of non-transfected parasites [13].” 

Materials and Methods

Line 299: please address which ELISA and PCR reactions were used in this work.

Response:  We indicated the competitive enzyme-linked immunosorbent assay and nested and added references.

Lines 316-317: “Holstein calves approximately 4 months of age and determined to be Babesia-free by RAP-1 competitive enzyme-linked immunosorbent assay and nested rap-1-PCR [6, 7],…”

Line 306: Why did the authors add this sequence? What does the appended chimera code for? Is it a short ORF, a tag?. Please explain in the text.

Response:  We added a statement regarding the purpose of adding the AM sequence.

Lines 330-331: “ The unique 108 bp sequence, that codes for 36 amino acids, was appended to allow dis-crimination between transfected and wildtype mRNA.”

Section 4.6. At what times after larval feeding did the authors verify the presence of parasites in the calves.? What acaricide was used and how was it applied?. Explain in the text.

Response:  We modified the statement for clarification.  We added acaricide information as requested.

Lines 401-404: “After terminating larval feeding using topical Permectrin II (Bayer Healthcare LLC, Shawnee Mission, KS), transmission was evaluated daily for 14 days using light microscopy of Giemsa-stained blood smears and wet mount slides by epifluorescence microscopy.”